

# Classification of botnet attacks in IoT smart factory using honeypot combined with machine learning

Seungjin Lee, Azween Abdullah, Nz Jhanjhi and Sh Kok

School of Computer Science & Engineering, Taylor's University, Subang Jaya, Selangor, Malaysia

## ABSTRACT

The Industrial Revolution 4.0 began with the breakthrough technological advances in 5G, and artificial intelligence has innovatively transformed the manufacturing industry from digitalization and automation to the new era of smart factories. A smart factory can do not only more than just produce products in a digital and automatic system, but also is able to optimize the production on its own by integrating production with process management, service distribution, and customized product requirement. A big challenge to the smart factory is to ensure that its network security can counteract with any cyber attacks such as botnet and Distributed Denial of Service, They are recognized to cause serious interruption in production, and consequently economic losses for company producers. Among many security solutions, botnet detection using honeypot has shown to be effective in some investigation studies. It is a method of detecting botnet attackers by intentionally creating a resource within the network with the purpose of closely monitoring and acquiring botnet attacking behaviors. For the first time, a proposed model of botnet detection was experimented by combing honeypot with machine learning to classify botnet attacks. A mimicking smart factory environment was created on IoT device hardware configuration. Experimental results showed that the model performance gave a high accuracy of above 96%, with very fast time taken of just 0.1 ms and false positive rate at 0.24127 using random forest algorithm with Weka machine learning program. Hence, the honeypot combined machine learning model in this study was proved to be highly feasible to apply in the security network of smart factory to detect botnet attacks.

## INTRODUCTION

The Industrial Revolution 4.0 has brought a great innovation to the conventional manufacturing into the new era of smart factories (*Oztemel & Gursev, 2020*).
The conventional factories involve automation or digitalization within each production process. This, however, make it very difficult to manage the entire production chain from general to specific levels. More innovatively, smart factory can effectively manage many processes in the production chain thanks to the use of many Internet of Things (IoT) devices. They are installed and interconnected with each other in every machine or equipment along the production chain. Hence, a smart factory is advantageous in

Corresponding author
Seungjin Lee,
leephael8707@gmail.com

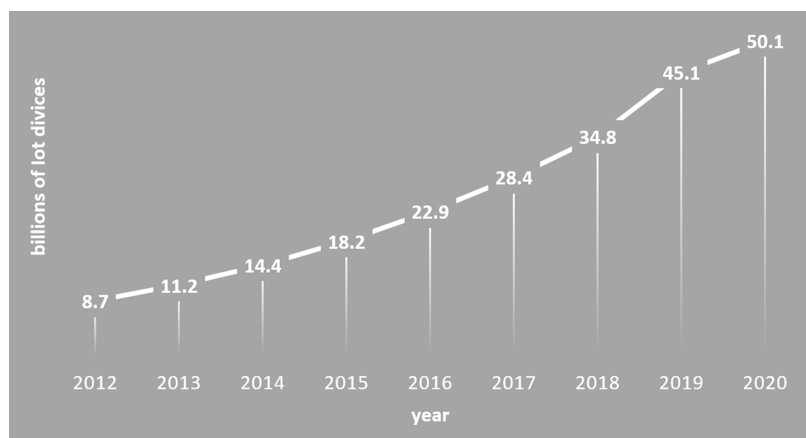

**Figure 1 Demand for IoT equipment.** The use of IoT equipment that is increasing every year.

producing a variety of products according to customer's desire at better quality and higher productivity. Also, IoT devices/equipment play a very important role in the operation and management of smart factories. A demand for the IoT equipment in smart factories has been increasing since 2012 as shown in Fig. 1. Especially in the last 5 years (2015–2020), the use of IoT devices has increased tremendously from 18.2 billion to 50 billion for application in the smart factories (*Smith, 2015*).

Additionally, as smart factories are combined with Information and Communications Technology (ICT), all the facilities and devices are connected at the central wireless communication. This allows data to be freely linked between the processes and provides a more systematic, integrated and optimal production environment. Efficiency in time management for production can be greatly enhanced with a minimal production cost. Therefore, products produced by smart factories become more competitive in the market.

Although IoT smart factories have been built and operated in the industry, standards of implementation for smart factories have yet to be established (*Guo et al., 2020*). Basically, a smart factory consists of three aspects, that is, interconnection, collaboration and execution, which all attribute to the manufacturing conceptualization of being adaptive and flexible (*Jiafu et al., 2016*). This concept is reflected in the architecture of the smart factory operating on IoT system as shown in Fig. 2 (*Chen et al., 2017*). With four layers arranged hierarchically, it starts at the physical resource layer, followed by the networking layer and the application layer, and ends at the terminal layer. A manufacturing system in the smart factory can be assessed from different layers (*Li et al., 2018*). With the aim of transforming conventional factories into smart factories, in-depth research needs to look into all layers.

From the security perspective, research should focus more on the physical resource/sensing layer, as it is directly related to the vast usage of the IoT devices in order to reinforce the security network for smart factories. Finding any security-related issues is one of the priorities required for a smooth system operation by means of resolving any

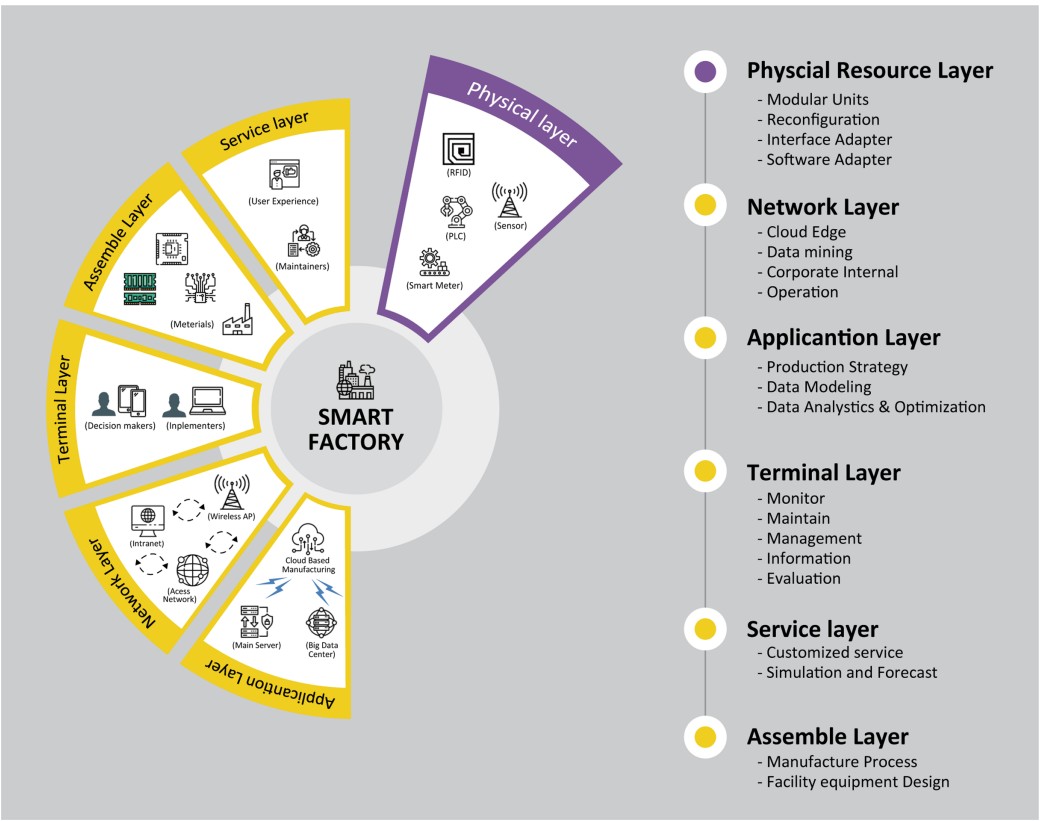

**Figure 2 Function requirements of smart factory.** Features for each layer are shown.

failover problems arising from the entire manufacturing chain (*Mittal et al., 2019*). Especially, the IoT devices are such as radio-frequency identification (RFID), CCTVs, programmable logic controller (PLC) equipment, sensor and main database servers are installed or located at the physical resource layer. Data transmission between these IoT devices can be easily affected in case that data leakage in smart factory network occurs. In the worst case, data updates can be abused by unauthorized users (*Ramos, Monge & Vidal, 2020*).

To mitigate the impact of data leakage and data abuse, real-time detection of cyber attacks to smart factory obviously becomes an extremely important factor to take into consideration of developing and improving security network of the smart factory (*Brett et al., 2009*).

Network security in the smart factory is highly at risk of being under cyber attacks due to the interconnection of a huge number of IoT equipment. According to a recent report, instability is recognized as one of the biggest limitations out of 250 vulnerable features found in the IoT devices (*Casalinuovo, 2019*). As a result, cyber attack to smart factories can easily spread to not only quality process control and production control, but also product design which can be analyzed or copied by unauthorized or illegal accesses. In the worse scenario, highly confidential information such as process know-how, requirements

of data analysis, product design drawings, R&D results are shared outside of the smart factory. Such threats of information leakage can cause serious damages and economic losses to both manufacturing and business sectors. This kind of cyber attack can be done by the act of exploiting security vulnerabilities in the ICT system via remote control or surveillance of systems in the IoT smart factory.

One of the most serious cyber attacks to smart factory is botnet. An example of the botnet attack is a temporary unavailability on some commercial websites such as Amazon, Netflix, Twitter, CNN and PayPal. A notable case ever recorded is the attack on the Dyn DNS infrastructure, which mobilized 100,000 IoT devices (mainly CCTV cameras) in October 2016. Another example is the new Mirai-source-code being launched in 2017. These Mirai-induced IoT botnets have occurred frequently in the recent years with a very serious consequence. Therefore, it is very important and urgent to identify and mitigate IoT botnets through the development of new technologies for network security (*Ozcelik, Chalabianloo & Gur, 2017*). As the number of attacks has soared due to unstable IoT devices in the Internet infrastructure, smart factories undoubtedly are highly possible to become an ideal victim of the IoT botnet attack.

Among many detection methods, honeypot has been investigated to apply for detecting botnet attack in various studies in the recent years (*Ja'fari et al., 2020*). However, a huge volume of attacking data collected by honeypot is highly complex and non-classified. This causes to lower the efficiency of botnet detection by the honey method in term of time taken and accuracy. In order to improve the efficiency, it is crucial to focus on classifying botnet attacking information and obtaining botnet attacking behaviors (intrusion type in other words). In this particular area of dealing with big data, artificial intelligence or machine learning has recently been applied effectively to speed up data processing, and make prediction as well as detection (*Seungjin, Abdullah & Jhanjhi, 2020*). Hence, it becomes very potential to apply machine learning for botnet classification, which is notably yet to be investigated in previous studies, especially in the smart factory environment.

Therefore, this study was aimed to investigate the feasibility of combining honeypot with machine learning in developing a botnet detection model for IoT smart factories. In this work, a configuration of hardware representing a physical layout of a smart factory was built and installed with software of honeypot combined machine learning. The whole setup was then programed to run for simulation to detect and classify botnet attacks into intrusion types.

Problem statement

The problem statement can be elaborated with following points:

- A huge amount of attacking data collected by honeypot is highly complex.
- Without data classification, efficiency of the honeypot model is low, since the current time taken is long but at low accuracy to detect botnet.
- A very limited number of studies focused on botnet detection for the smart factory.
- Strategy of using honeypot with machine learning has been suggested very recently with only study framework and lack of model verification for supporting.

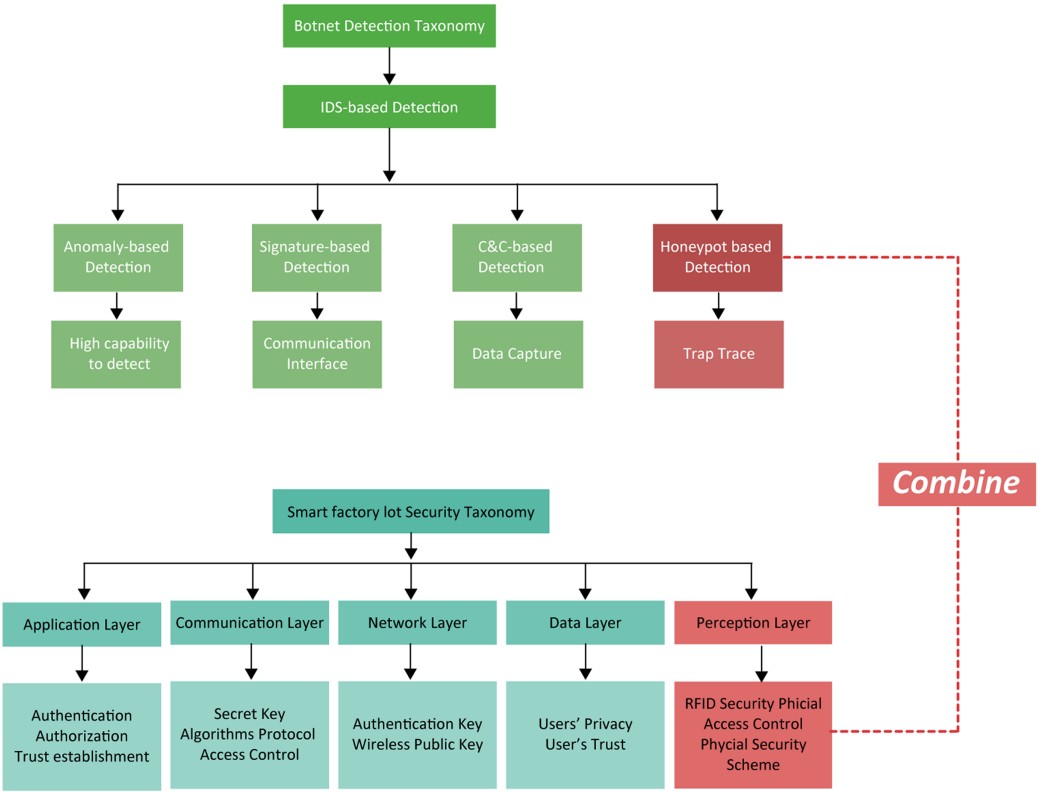

**Figure 3 Taxonomies for botnet detection and security layers of IoT smart factories.** The combination of honeypot detection with the perception layer is shown.

Study approach

- Apply machine learning as a supporting tool for classifying botnet attacks captured in log files generated by honeypot. Selection of random forest algorithm for machine learning to improve the classification process. Model testing on a hardware configuration mimicking a real smart factory environment.

## Related work

Various botnet detection methods and their rationale are described in the Taxonomy in Fig. 3. Besides, a smart factory is layered into the perception (physical), communication, network, data and applications with the function of each, as shown in the following taxonomy of the same figure.

Comparison of various detection models are presented in Table 1. Real-time detection is a very important factor which smart factories seek for *Katz, Piantanida & Debbah (2017)*.

- Honeypot can respond to attacks in real time and attract attackers to deceptive assets instead of actual assets (*Duessel et al., 2017*). Whereas for binary, anomaly detection methods, the response to real-time is however slower than that of the honeypot method (*Gerstmayer et al., 2017*; *Fenzl et al., 2020*).

- Binary detection is simple in the structure of using 1 and 0. But its detection processing speed is too low to be compatible to smart manufacturing environment which seeks real-time detection (*Katz, Piantanida & Debbah, 2017*)
- Although the command and control (C&C) method targets for HTTP-based botnet detection and expansion, its structure is not simple to implement and the detection result is high false-positive (*Fedynyshyn, Chuah & Tan, 2011*).
- In terms of cost effectiveness, honeypot has an advantage since it requires a relative low cost of construction and management (*Aziz, 2011*).

With an increasing interest in the potential application of machine learning, it offers a new solution for detecting abnormalities in the malicious Internet traffic. In fact, the Internet traffic which allows communication between IoT devices is distinguished from the Internet connectivity which runs on a variety of web server Many Internet-connected devices are such as smartphones, computer, laptops using a variety of web servers. Moreover, for the IoT devices, patterns of the network traffic are repeated in the regularity of network ping with small packets for logging.

Applying machine learning in botnet detection for smart factories can become useful to enhance performance of the honeypot model in term of speeding up the processing time or detection time (*Lim et al., 2019*). Interestingly, there have been very few studies making attempts to mount both honeypot and machine learning on IoT device networks to target attacks on the IoT traffic.

Table 2 summarizes a few studies in botnet detection using the approaches of honeypot and/or machine learning.

- IoT botnet detection is an approach used to design a detection model based on the binary when botnet attacks IoT device as a hypothesis (*Choi, Yang & Kwak, 2018*). Although monitoring algorithms for the infected IoT device are simple and easy through web services, capacity of the IoT devices has certain limits as a restriction in the IoT botnet detection.
- Another approach to detect botnet is using machine learning which gave a high accuracy in detection at 91.66% (*Wang et al., 2020*). However, one disadvantage of using machine learning approach is that fast detection is hard to achieve in the randomized number of packets. Consequently, the feasibility of applying this approach for smart manufacturing needs more research looking into real-time factor and accuracy.
- Botnet detection using honeypot integrated with IoT, named as IoT honeypot was studied in the environment of smart factories (*Dowling, Schukat & Melvin, 2017*). In comparison with the machine learning approach by *Wang et al. (2020)*, the IoT honeypot approach is able to gather information at high speed with less resource consumption (*Jiafu et al., 2016*). Although the IoT honeypot approach has been shown to be scalable by applying it to sandboxes IoT to support high protocols, more expansion in various situations and environments is needed with features to activate the architecture of the IoT devices (*Jiafu et al., 2016*).

- One study suggested to apply machine learning to detect botnet in the smart factory environment (*Park, Li & Hong, 2018*). On the one hand, the machine learning approach can reduce the cost as an advantage. On the other hand, low detection rate and high complexity and uncertainty are recognized as big limitations. Thus, it might not suitable for smart factories, unless a construction of machine learning with Kenta-aware intrusion tower system is built, which bears an additional cost.

- Advancing from the IoT honeypot approach and the machine learning approach, honeypot combined with machine learning named as honeypot machine learning uses learning logging for detection and tracking at high accuracy (*Vishwakarma, 2019*). In accordance with most standard equipment at various functions, the honeypot machine learning approach is suitable for the performance of smart factory with a minimal resource required. Hence, it is likely to be adopted in the future (*Vishwakarma, 2019*).

Among those approaches being discussed, IoT botnet and honeypot machine learning approaches shows some effective results in detecting botnets. These two approaches are possible to trace through logging at low cost and are most cost-saving for the IoT devices. Notably, feasibility of applying the honeypot machine learning approach in the smart factory especially has yet to investigate in any studies so far as being highlighted in the recent review work (*Seungjin, Abdullah & Jhanjhi, 2020*). Therefore, more research in botnet detection should look insight into this particular area for application in smart factory.

## MATERIALS AND METHODS

### Proposed model

This section is organized to focus on three main aspects. The first and second aspects mention configuration and simulation of hardware in a virtual smart factory environment. The last aspect presents algorithms of the honeypot detection model in combination with machine learning programing.

### Configuration of hardware for a virtual smart factory environment

In the configuration setup, some IoT devices (camera, RFID, temperature sensors) and two raspberry pie devices (Pi1 and Pi2) were used to create a virtual smart factory environment.

The first Raspberry Pi (Pi1) was assigned as the actual main IoT data collection server by installing Open CVS. It was responsible for transmitting collected IoT data to the main PC. T-pot platform was chosen because it was suitable for virtual experiments using raspberry and allowed to monitor real-time botnet detection through dashboards. The second raspberry pi (Pi2) was installed with a virtual server (VM) (T-pot platform) So that collection of detection information in such the environment was deliberately established. One assumption was that botnet attacked the raspberry pi 2 (honeypot server) using 10 feature botnet datasets. Raspberry Pi1 and Pi2 were installed on a log server to

**Table 1 Comparison of honeypot with other detection methods.**

|  | Honeypot | Binary detection | Anomaly detection | C&C detection |
|---|---|---|---|---|
| Configuration | High-interaction virtual server | Binary | P2P | Command & control server |
| Advantages | Monitor the interaction of the grid with infected devices User friendly UI system | Easily applicable to multi-connection systems | Systems have the capability to detect zero-day attacks as well | Able to detect and expand HTTP-based botnets. |
| Disadvantages | The analysis of information on an attack is slow and passive | Spend a lot of time training | Not simple structure high false-positive | Not simple structure High false-positive |
|  | (*Zhang et al., 2019*; *Duessel et al., 2017*) | (*Gerstmayer et al., 2017*) | (*Fenzl et al., 2020*) | (*Fedynyshyn, Chuah & Tan, 2011*) |

**Table 2 A comparison of studies in botnet detection using honeypot and/or machine learning approaches for smart factories.**

| Approaches | Strengths | Weaknesses | Research gap | Refs. |
|---|---|---|---|---|
| Machine learning for smart factory environment | Cost reduction | Detection rate is very low and the accuracy is low. The system is complicated | Building machine learning and Kenta-aware intrusion tower systems for information that will be leaked from manufacturing processes | *Park, Li & Hong (2018)* |
| IoT Botnet | Monitoring Web-based real-time IoT equipment. Easy and simple interface | Limited capacity | New optimization requires expansion of utilization | *Choi, Yang & Kwak (2018)* |
| Machine learning | 91.66% graph-based detection accuracy | Difficult to apply flow-based detectors | A graph-based bot mark is required to increase the accuracy of botnet detection | *Wang et al. (2020)* |
| IoT Honeypot | Speed of gathering information is fast. Less resource consumption. | Unnecessary data piles up | It is necessary to activate network protocol by expanding IoT equipment and sandbox | *Jiafu et al. (2016)* |
| Honeypot machine learning | Real-time monitoring with the combination of honeypot and machine learning | It is greatly affected by the system environment | Problems with device data capacity cloud server application | *Vishwakarma (2019)* |

keep the information on IoT product line in the factory and records of botnet attack pattern time zone, making it easy to track.

Operating mechanism of the honeypot combined machine learning model in smart factory is illustrated in Fig. 4. When an attacker attempted to inject a malicious code through an open port. This step was done by logging into an IoT device at the physical resource/perception layer by combining multiple IDs and passwords. The honeypot intentionally broke into his protective wall and came in as a person who could reach the attacker. The main intention was to obtain information about attackers and malicious code botnets by recording each activity between the device and the intruder in the form of a log file. These log files captured information that allowed administrators to identify characteristics, transformations, target device types, C&C server IP addresses, port numbers of the new malware suites or botnets. Log file data was then converted to an appropriate table format that can be used as a dataset.

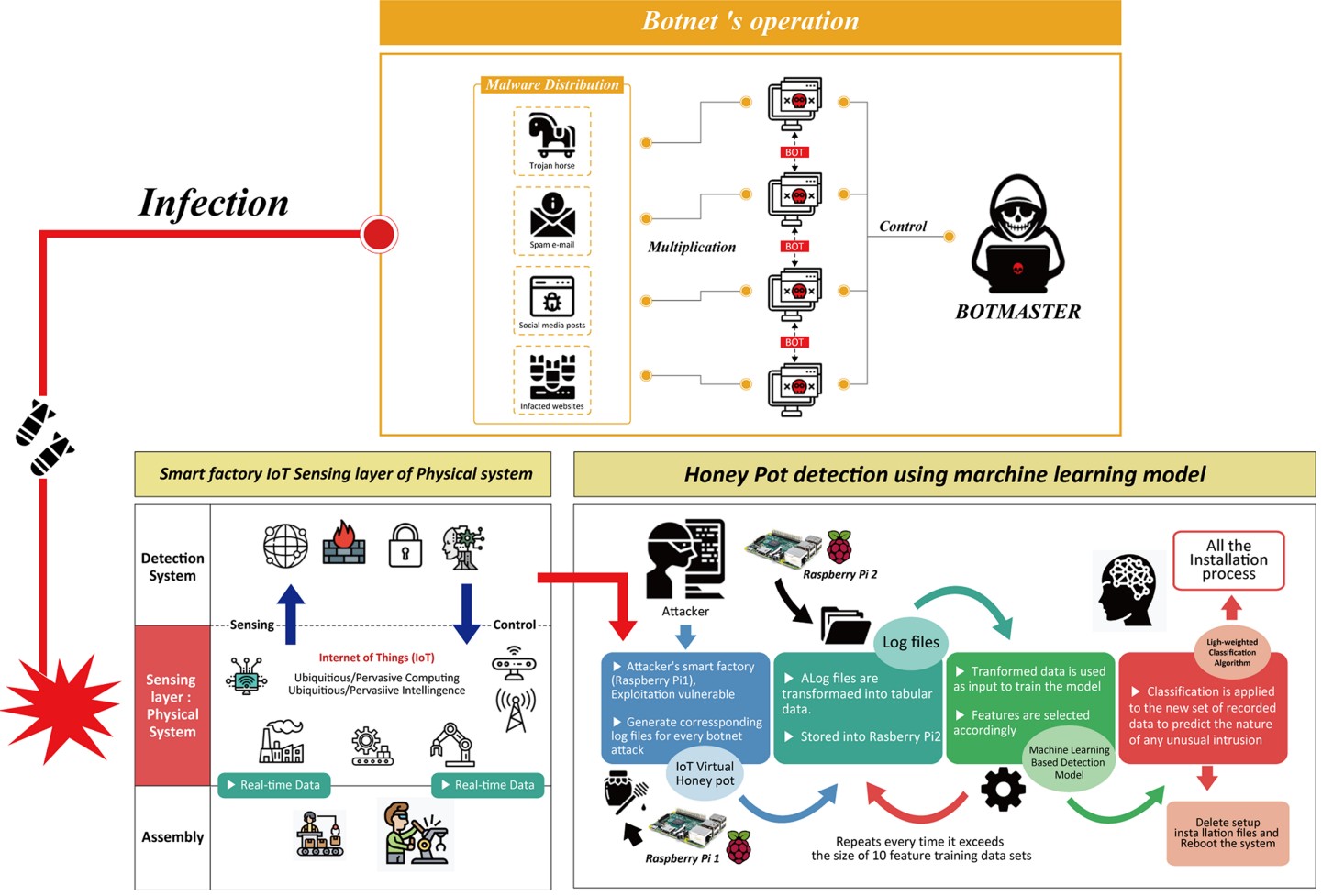

**Figure 4 Operating mechanisms of botnet attacks to the physical sensing layer of a smart factory and design of the honeypot combined machine learning model.** Botnet detection by the honeypot at the physical layer is shown.

A machine learning (ML) tool was then trained with the dataset which contained 10 network parameters. These parameters were based on the most 10 common features of IoT botnet attacks to smart factory reported in the previous work studying in network architecture and network type used in the physical layer of smart factory, smart home network and smart city (*Fan et al., 2020*; *Almusaylim & Zaman, 2019*; *Humayun et al., 2019*).

Algorithm written for this ML tool classified botnet data using R-studio and Weka. Memory-efficient classification was desirable to predict useful information by using less training data to prevent IoT devices from becoming overwhelming. Afterwards, appropriate measures were taken according to the results of the classification. Whenever the course exceeded the allowed size of training data, it would dynamically repeat.

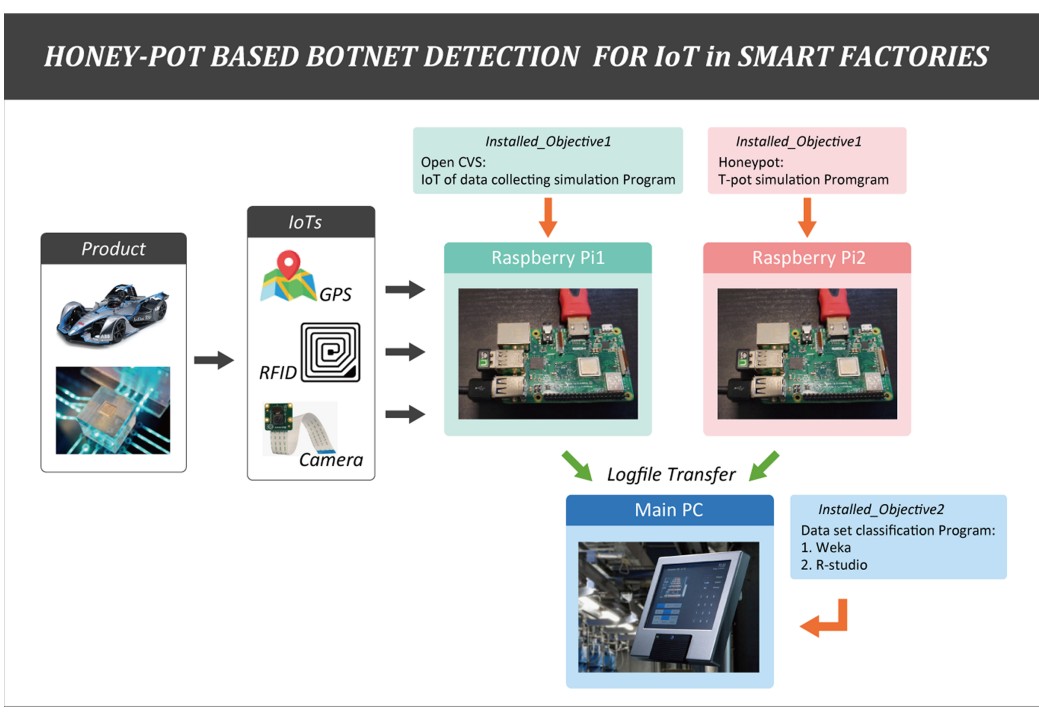

**Figure 5 Simulation of hardware design.** The collection of product information by Raspberry Pi is shown.

## Simulation of hardware design with raspberry Pi transfer log files

Each product in the smart factory was attached with a Radio-frequency identification (RFID) tag containing information. Camera reads the RFID tags to collect product information. The collected information was then stored in the raspberry Pi1 and Pi2 as the calling terminals. In other words, it was programed to transmit the collected data to the IoT service server of the network through a terminal (camera, temperature, and RFIDIoT devices). The information kept in Pi1 and Pi2 as a log file was then transferred to the server of the main PC. This process simulation is illustrated in Fig. 5.

## Raspberry Pi1 setting: data transporting Open CV

Transporting and receiving of data in this virtual environment were similar to those taking place in the real smart factory. Data transporting Open CV was used as a collection of Python classes that transferred Open CV images and data from the raspberry Pi1 to the main computer via Data transporting Open CV messages.

For example, on the main computer screen, video and picture streams were shown simultaneously by sending signal data of raspberry Pi1 as shown in Fig. 6. Algorithm was required for the main PC and raspberry Pi1 for such data transfer as shown in Table 3.

## Raspberry Pi2 setting: T-pot honeypot platform

After Raspberry Pi1 setting with data transporting Open CV, Raspberry Pi2 was set with T-pot honeypot platform followed by virtual machine (VM). Verification and testing were

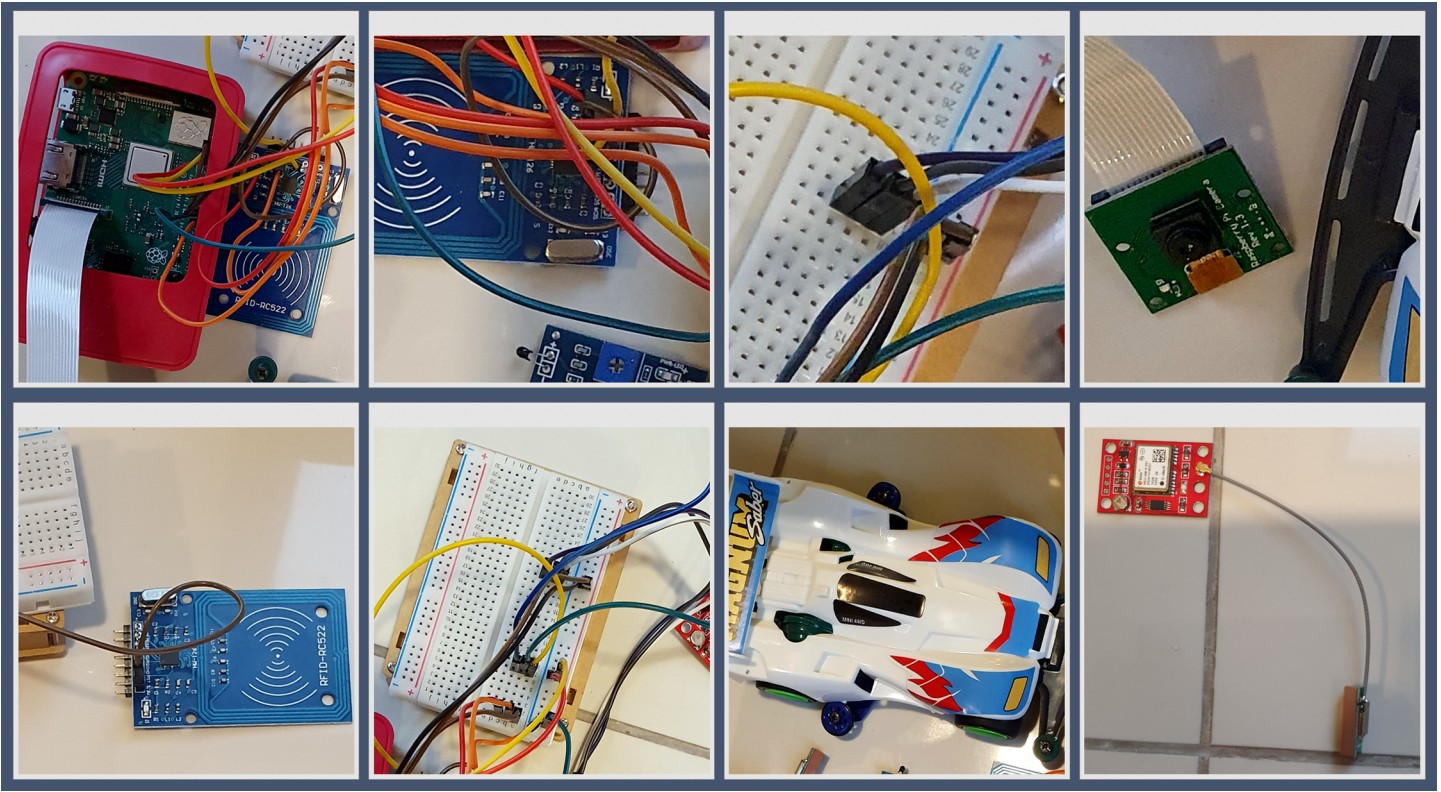

**Figure 6   Raspberry pi1 image transfer data.** Captured data by IoT camera.

done on all the honeypot to be balanced at runtime. To do this task, a studio state command line was used to write script and verify the transmission of log file.

The script in Fig. 7 shows the load on the platform, the status of each honeypot, and the uptime. Furthermore, the data collected by the honeypot was visually displayed using the Kibana dashboard showing network attacked by malicious users and botnet. The Kibana dashboard shown in Fig. 8 is convenient and comprehensive for analysis of the type, location, and malicious threat actors of botnet attacks. It infiltrated the Raspberry Pi server inside the VM. This had many potential uses for data systems and metrif collection in smart manufacturing environment that required real-time monitoring.

## Honeypot and machine learning classification process design and algorithms

Diagram in Fig. 9 describes the process design for botnet detection by honeypot combined with machine learning classification model. The entire process consisted of two stages. In the first stage (honeypot simulation), it took place at the raspberry Pi2 server which finished its loading by checking botnet credentials and then started the honeypot detection in the T-pot platform.

To verify botnet credentials in this step, a user name and password must be provided. Accurate information given by authorized users would be proceeded to start honeypot detection. The verification step took approximately 60 s.

**Table 3 Algorithm of data transfer in raspberry Pi 1.**

**Algorithm: Raspberry Pi image data transfer**

1. **Input:** List of data transfer
2.     **Output:** PI image data (task mapped to VM)
3.     **Begin**
4.     **True:** CV, VM. Show streamed images and data
5.       **If**
6.         two tasks get data from RFID, image, signal then
7.         Pick task with earliest
8.     **Else**
9.       Transfer data
10.     **while VM**
11.       Compute Utilization
12.       Sent image, signal each
13.     **Repeat** If data is available & task allocated then
14.       Migrate task to less utilized data
15.     **Else**
16.       Start scheduling
17.     **Until** all send images as stream
18.       Image = Pi Cam read
19. **End**

**Figure 7 T-pot test script for raspberry Pi2.** User accounts checking in T-Pot.

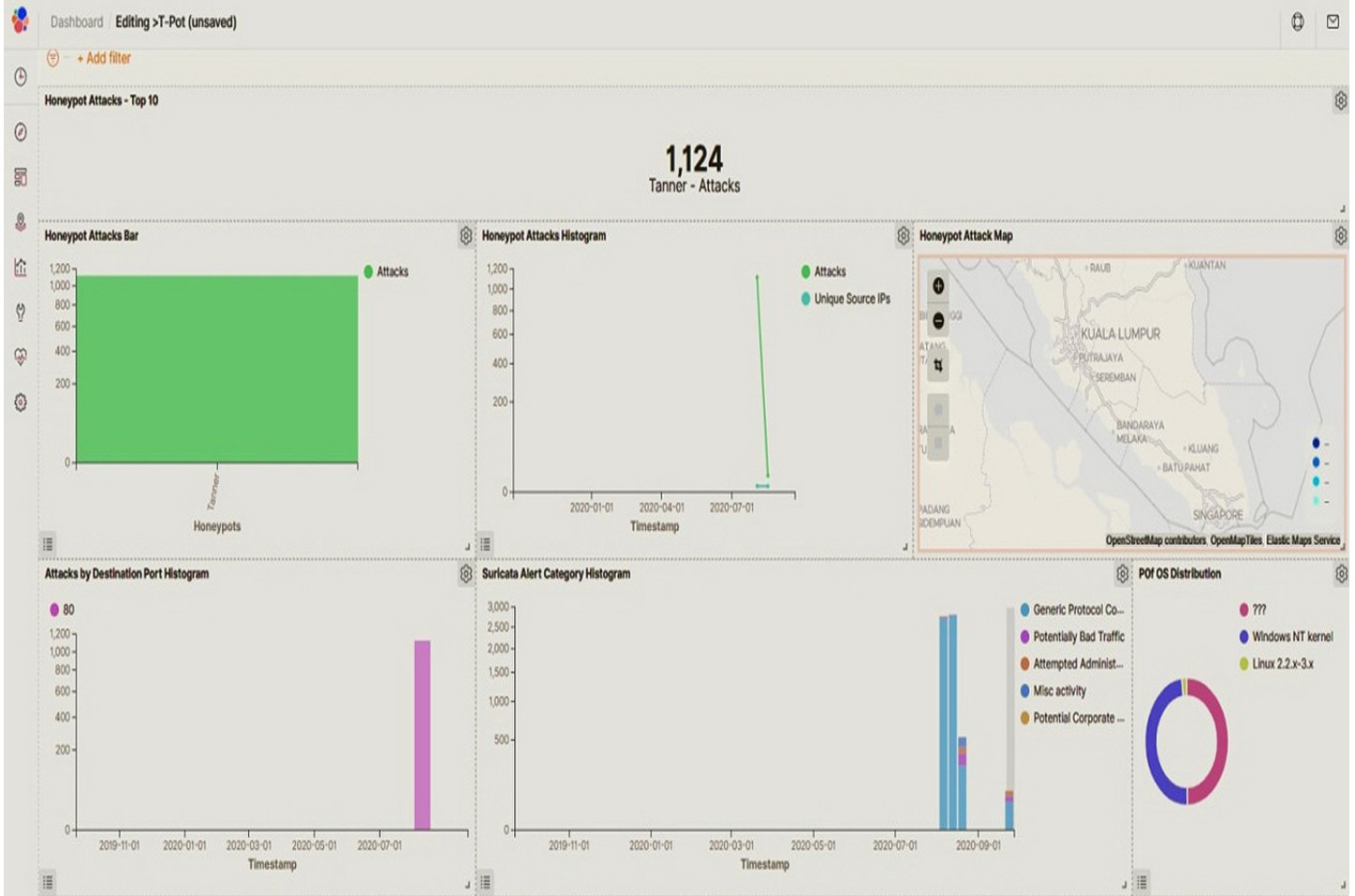

**Figure 8 Kibana dashboard.** Metrics are shown by integrating detection real-time threat charts, maps, and filters.

Moving to stage 2—log data collect, data which was earlier obtained from the authorized users in stage 1, was being processed through a series of steps, that is, retrieve logs, extract records, filter, text processing, upload, end automated process. During the text processing step before uploading, this was where machine learning was integrated with the honeypot for the purpose of classifying botnet attacks as illustrated in Fig. 9.

As a result of classification, botnet attacking behaviors or botnet intrusion types were obtained and therefore used for machine learning training to detect botnets. After machine learning classification, processed data was converted to output result file and uploaded to end the automatic process. Algorithm for botnet classification by machine learning is shown in Table 4.

Instruction to verify the codes and dataset:

1. Read the package setting for CARET, DPLYR, and READR in the library (this package can balance the dataset for four categories of botnet attacks).
2. Make setting in the computer for dataset with frame set and honeypot log file.

3. Start the test based on the 10 best feature data and wait until the true setting to come out.

4. Proceed to filter, extract records, review logs and test processing.

5. Apply (flgs_number, srate, drate, rate, max, state_number, mean, min, stddev, seq) to each classified category.

6. Obtain log files as samples for machine learning classifying into four types of botnets (Distributed Denial of Service (DDos), Dos, Reconnaissance and Theft).

7. Call in the algorithm (random forest or SVM) to start classification.

8. Predict botnets based on the classification result in term of accuracy, time taken, false positive rate, and $p$-value.

## RESULTS

In this section, results were collected mainly from the machine learning classification of botnet attacks. The data collection for the experiment was randomized if the reference was used. The experiment was carried out using Raspberry Pi and a personal computer. Features of the selected dataset were first described in Dataset section. To perform the classification on the dataset, honeypot combined machine learning model simulation was run by two techniques that is, Weka, and Rstudio. The collected results were evaluated by comparison between the two, and with other study.

### Dataset

In order to use a machine learning method to identify botnet as the target of IoT-based network and physical layer in smart factories, we experimented with data set 10 features of this paper (*Koroniotis et al., 2019*), which is the most suitable data set for this study.

Dataset used for classification was selected based on ten features which were closely related to the botnet intrusion types. Details of the 10 features are presented in Table 5, which was extracted from a direct comparison of entropy and correlation scores in the previous study (*Zheng & Keong, 2011*, *Koroniotis et al., 2019*). Specifically, transmission formation was calculated as correlation indices which were evaluated for their statistical measurement values.

The calculation of indices was performed using the following equation.

$$x_i = (x_i - x_{\min}) * \frac{(b - a)}{(x_{\max} - x_{\min})} + a$$

Model simulation was evaluated using some of the evaluation metrics of machine learning as shown in Table 6 (*Koroniotis et al., 2019*). Based on the results, the model can be evaluated whether it is highly efficient in optimization and able to reduce the error of drate. The values of max and min were to represent the values of training and response. Examples of correlation are in Fig. 10 (*Koroniotis et al., 2019*).

### Classification by Weka-machine learning

Classification result of the Weka-machine learning technique for four types of botnet attacks namely DDoS, DoS, reconnaissance, and theft is shown in Fig. 11. For 76 instances,

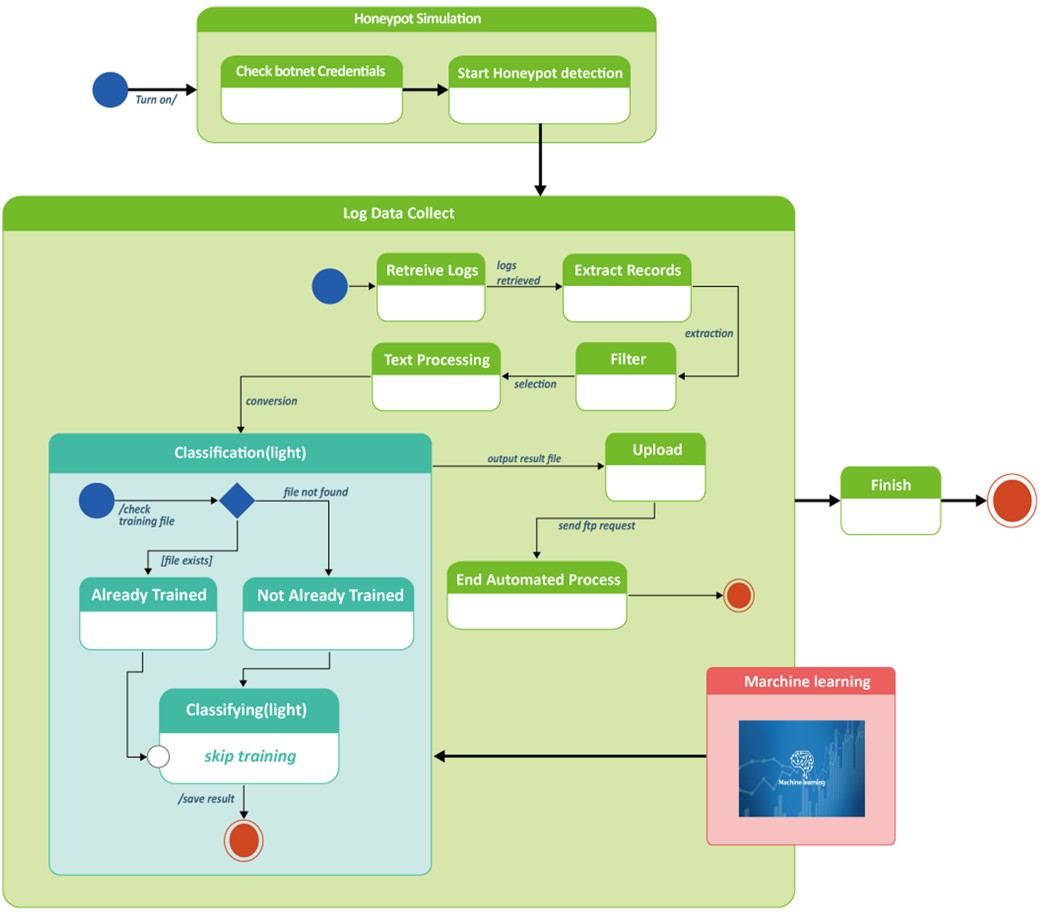

**Figure 9 Process design of honeypot combined machine learning model to detect botnet attacks in smart factory.** Honeypot simulation, log data collect classification flowchart.

the average percentage of correct prediction (accuracy) for all four botnet attacks achieved 96.00526%. The kappa statistic showed the model stability was 0.9466 with a mean absolute error of just 0.0478. In term of accuracy and precision for each type of attacks, reconnaissance has the highest values.

Furthermore, after collecting pcap files from the virtual settings, statistical measures using correlation coefficients and entropy techniques were adopted to extract flow data using Argus tools in order to evaluate datasets based on the 10 best features. A new function was created based on the transaction flow of network connections to find out normal and intrusive events. A machine learning model has been trained and validated in different versions of datasets to assess the value of datasets compared to other datasets as shown in Fig. 12.

## Classification by Rstudio-machine learning

Figure 13 shows the results of classification using Rstudio machine learning technique. Two methods were used, that is, support-vector machine (SVM) and random forest (RF). RF was calculated using the decision tree (DT) to predict mean values. RF was selected

| Table 4 Algorithm for data classification. |
| --- |
| **Algorithm: botnet-based classification ML and honeypot detection for improving the security of smart factory IoT** |
| 1. **Input:** List of Classified 10 feature-valued training data set (t1, t2, t3, …, tn) & VM |
| 2.  **Output:** 10 feature tasks DT |
| 3.  **Begin** |
| 4.  **Initialization:** DT; 1.Sort tasks according to DT ascendingly |
| 5.    **If** |
| 6.    DT belongs to same VM |
| 7.    TA = testing attribute |
| 8.    {combine DT =T; |
| 9.     **Else** |
| 10.    Priorities based on (Data set ) |
| 11.     **For each VM** $i$ = to $n$ |
| 12.    {**calculate Information_gain**} |
| 13.     **Repeat** If change is available factor & TA **then** |
| 14.  **For** (Each DF in the splitting of TA) |
| 15.  **If** (T' is empty) |
| 16.  **Else** |
| 17.  Start get sample again |
| 18.   **Until** all 10 features allocated to a VM |
| 19.  **End** |

| Table 5 Ten Features of the selected dataset for classification. | | |
| --- | --- | --- |
| No. | Name of features | Description |
| 1 | srate | Foundation-to-target t time packets for each second |
| 2 | drate | Target-to-foundation packets for each second |
| 3 | rate | Over-all packets for each second in transaction |
| 4 | max | Maximum period of collected archives Source |
| 5 | state_number | Numerical illustration of characteristic state |
| 6 | mean | Average period of collecting records |
| 7 | min | Minimum time of collecting records |
| 8 | stddev | Standard deviation of aggregated records Total |
| 9 | flgs_number | Numerical representation of feature flags |
| 10 | seq | Argus sequence number |

because it showed the effective detection in discrete datasets such as botnet (*Kok, Abdullah & Jhanjhi, 2020*). Evaluation of the classification result were based on nine parameters namely, sensitivity, specificity, pos pred value, neg prered value, prevalence, detection rate, detection prevalence, balance accuracy, average.

In the RF method, a high accuracy was achieved at 0.96. The 96% CIs were 0.8875 and 0.9917. The RF provides Kappa at 0.964. However, for SVM method, the accuracy was obtained at 0.7733 which is much lower than that of the RF. The 96% CIs were 0.6666,

**Table 6 Machine learning evaluation metrics.**

| | |
|---|---|
| Accuracy | $\text{ACC} = \dfrac{\text{TP}}{\text{TP} + \text{FP}}$ |
| Precision | $\text{PPV} = \dfrac{\text{TP}}{\text{TP} + \text{FP}}$ |
| Recall | $\text{TPR} = \dfrac{\text{TP}}{\text{TP} + \text{FN}}$ |
| Fall-out | $\text{FPT} = \dfrac{\text{FP}}{\text{FP} + \text{TN}}$ |

**Notes:**
TP (True Positive): number of botnet containers represent as botnet.
FP (False Positive): number of regular containers symbolized as botnet.
TN (True Negative): number of normal containers represent as standard traffic.
FN (False Negative): number of botnet containers symbolized as standard traffic.

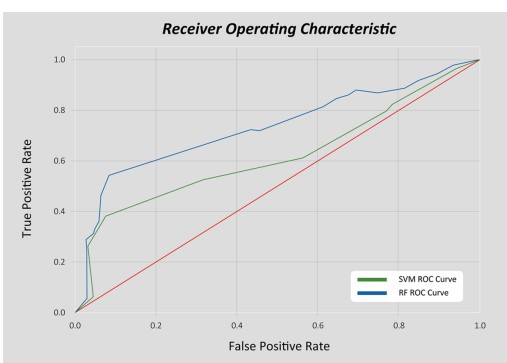 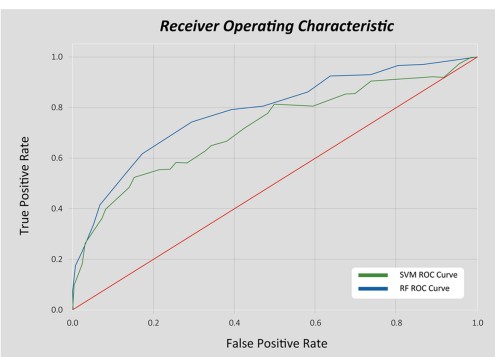

A) ROC curve for RF,SVM Model (area = 0.701)       B) ROC curve for RF,SVM Model (area = 0.976)

**Figure 10 Correlation of true vs. false positive rates for the 10 best features.** (A) ROC curve for RF, SVM Model (area = 0.701); (B) ROC curve for RF, SVM Model (area = 0.976).

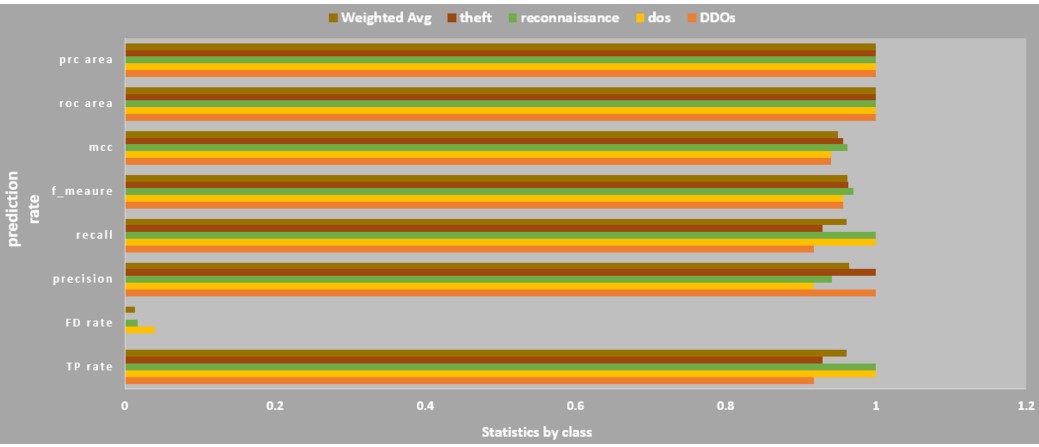

**Figure 11 Detailed accuracy by classification.** The estimated trend for each botnet type is shown.

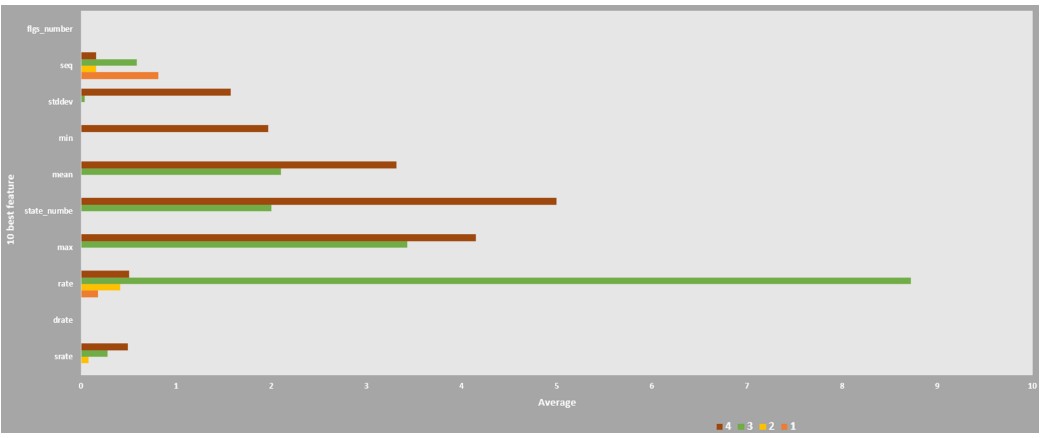

**Figure 12 Correlation of 10 features for Weka-machine learning.** Training and Testing Classification result is shown.

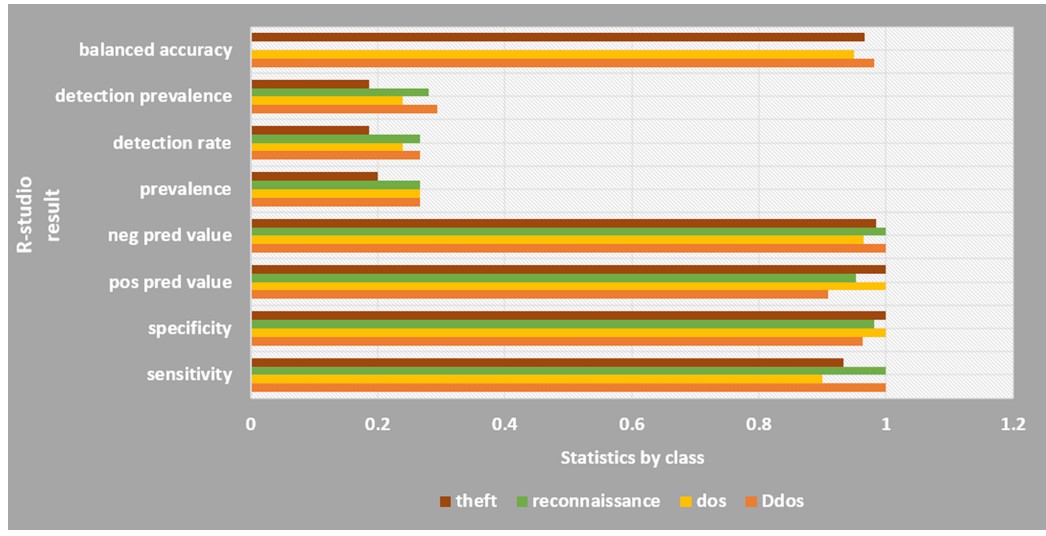

**Figure 13 R-studio results.** The DDoS attack type is highly aggressive.

0.8621. As can be seen in Fig. 13, classification of DDoS attack type is highly efficient using Rstudio with respect to all 9 evaluating parameters. The detection rate and detection prevalence have low probabilities of 0.24 and 0.25 respectively. The overall detection using R-studio showed a statistically significant result since $p$-value is less than 5%.

## DISCUSSION

The two machine learning programs (Weka and Rstudio) following the random forest algorithm showed good result of classification and comparable with another study as shown in Fig. 14. In the study of *Mathur, Raheja & Ahlawat (2018)*, botnet were detected via mining of the network traffic flow with random committee method. The resultant accuracy of the random committee was achieved at 95.3%, which was 1.3% lower than

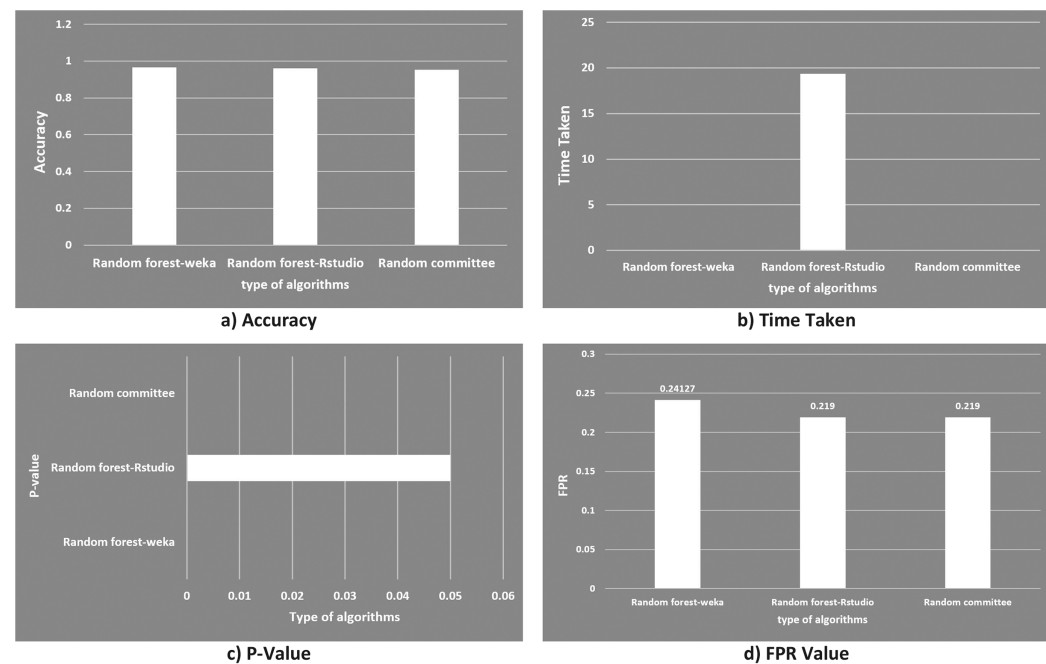

**Figure 14 Graphical comparison of random forest in this study with random committee in other study.** (A) Accuracy; (B) time taken; (C) *p*-value and (D) FPR value.

those obtained in this study at 96.66667% for random forest-Weka and 96% for random forest-Rstudio. In term of time taken, both random forest-Weka and random committee were able to detect botnets within a very short time of 0.00001 s or 0.1 ms. Whereas it took a very much longer time of 19.37 s for the random forest-Rstudio to detect botnets because of its program package. In addition, the *p*-value representing the significance of the detection models, p value to the random forest-Rstudio was less than 5% ($2.2 \times 10^{-16}$), which showed that the detection model is statistically significant. As mentioned earlier that real time detection is the key factor importance to the network security of the smart factory, random forest method is therefore considered to be highly suitable for the environment of smart factory operating 24/7 in time.

The result of this study can be said to be just relatively comparable with the work of *Mathur, Raheja & Ahlawat (2018)*, since both of the studies were based on network traffic flow and targeting at the botnet attack. However, for the feasibility to apply in the smart factory environment, this study shows to be more feasible because of two reasons. First, the experiments were conducted on simulation of hardware specifically configured to mimic the real smart factory environment using IoT devices as mentioned in the proposed model section, which is lacked in the work of *Mathur, Raheja & Ahlawat (2018)*. Secondly, the experimental results obtained in this study addressed directly to the three deciding factors (time taken: 0.1 ms, accuracy: above 96% and FPR: 0.24127, refer to Table 7) which are very useful for evaluating any tested methods being applied in the smart factory.

**Table 7 Result comparison between random forest with random committee algorithms.**

| | Accuracy (%) | Time taken (detection time) | Fall positive rate (FPR) | *p*-value | Refs. |
|---|---|---|---|---|---|
| Random forest-Weka | 96.66667 | 0.0001 s | 0.24127 | – | This study |
| Random forest-Rstudio | 96 | 19.37 s | 0.219 | 0.05 | |
| Random committee | 95.3 | 0.0001 s | 0.219 | – | *Mathur, Raheja & Ahlawat (2018)* |

Whereas the latest publication work in 2020 reported that their study was based on smart factory ambient/environment to detect context-aware intrusion using machine learning (*Park, Li & Hong, 2020*). But there was no mention in that study to target any specific types of cyber attacks or virus, but only for anomaly signs. Another limitation of the study is that its result just stated a very general possibility of process improvement of 29% from 1.29% (*Park, Li & Hong, 2020*). Without showing the three deciding factors (time taken, accuracy and FPR), it can be hardly possible to evaluate the feasibility to apply for the smart factory. In addition to this point, the part of using machine learning used for training to obtain intrusions did not mention to include the best features of smart factory in the datasets. If including it would be very helpful to increase the feasibility or applicability of any detection models at the physical layers with interconnection of many IoT devices, as this present study were conducted accordingly. Furthermore, the Kibana platform supporting for a visualization of the system/model performance could provide a user-friendly interface for the administrators in the smart factory to analyze from a variety of perspectives more than just a visible display.

This study had provided a basic background for developing a security network just for the smart factory environment with a mimicking IoT device hardware configuration and random algorithms for the experimental work. The results can be used as reference points or benchmarks for more comparison with other future studies relating to smart factory. In fact, the number of studies focusing on the security network for smart factory to target specifically botnet attacks using honeypot is currently scarce throughout the literature, future work can based on this smart factory hardware configuration design for the experimental testing for models or systems. Also, it is suggested to further this study by conducting experiments in a real smart factory. By doing that, a better result can be obtained for analysis when many factors of smart factories are taken into consideration. Instead, a controlled virtual smart factory environment was created in this study. The expected results will be more valuable for improving the productivity of smart factories.

## CONCLUSIONS

In this work, the model of combining honeypot with machine learning was proved to be feasible in detecting botnet in the smart factory. Since the botnet can be easily spread into IoT smart factory environment with a high risk, hardware-based simulation and classification using random forest algorithm for Weka machine learning program showed

a very good result. 96.66667% for accuracy, 0.1 ms were achieved for the proposed model to detect botnet and the FPR was low at just 0.24127. Comparing this result to other studies showed that the proposed model (honeypot combined machine learning to detect and classify botnet attack) in the smart factory was evaluated to be better because of three outstanding advantages. First, IoT devices were used in the hardware simulation configured to mimic the real smart factory environment. Second, the result of model testing has showed that with a short time taken: 0.1 ms, high accuracy: above 96% and low FPR: 0.24127 by the random forest Weka machine learning as the deciding factors. Lastly, machine learning have been used the dataset which included the best 10 features of the smart factory for training to obtain intrusions.

# ACKNOWLEDGEMENTS

The author acknowledge the support of Taylor's University, School of Computer Science and Engineer in carrying out this experimental research work. This research work has been underwent for English proofreading prior to submitting to this journal.

## Funding
The authors received no funding for this work.

## Competing Interests
The authors declare that they have no competing interests.

## Author Contributions
- Seungjin Lee conceived and designed the experiments, performed the experiments, analyzed the data, performed the computation work, prepared figures and/or tables, authored or reviewed drafts of the paper, and approved the final draft.
- Azween Abdullah analyzed the data, authored or reviewed drafts of the paper, and approved the final draft.
- Nz Jhanjhi analyzed the data, authored or reviewed drafts of the paper, and approved the final draft.
- Sh Kok performed the computation work, authored or reviewed drafts of the paper, and approved the final draft.

## Data Availability
Raw data is available at *Koroniotis et al. (2019)*. Code is available in the Supplemental Files.

## Supplemental Information
Supplemental information for this article can be found online at http://dx.doi.org/10.7717/peerj-cs.350#supplemental-information.

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
