# Peer review of "Classification of botnet attacks in IoT smart factory using honeypot combined with machine learning"

_PeerJ Computer Science, doi:10.7717/peerj-cs.350_

## Round 0.1 · original submission · Major Revisions

Please revise the manuscript based on the comments of the reviewers.

Reviewer 1 ·

Basic reporting

In this paper, the authors presented an experiment where honeypot is combined with machine learning to detect botnet in the smart factory.

The paper well is well written in terms of English, but the overall structure of the paper is not well organised. The presentation of the paper needs to be improved for better understanding of readers.

Instead of writing your views in the form of long sentences, authors should follow the following points:

1. Better to use more bullet points to discuss problem and solutions separately.
2. Use a preliminary section, where describe several terms/definitions used throughout the paper.
3. Use tables in the related work so that reader can easily compare the state of art work results. No one wants to read long sentences to compare the results of the paper.
4. Try to draw some figures to explain your point of views and several attacks.

Overall, please improve to presentation of the paper to make it more interesting.

Experimental design

Authors uploaded files related to code of the experiment. However, I suggest to upload them on github and also mention some instructions to implement them like standard projects. However, if not willing to publicly share the code in such case authors should upload some instruction to verify the codes and dataset.

Validity of the findings

Results presented by authors are naive and can be verified.

Additional comments

Please improve the overall presentation of the paper as mentioned above.

Reviewer 2 ·

Basic reporting

Authors proposed a honeypot combined machine learning model that is proved to be highly feasible for detecting botnet in the smart factory. The paper is interesting, and the work itself is novel.

However, I strongly recommend authors to work properly on the presentation of the paper.

Experimental design

The experimental design of the paper is good, and it presents important results.

Validity of the findings

All the results can be verified from the available code and therefore, no issue with the validity of the article. However, pls also upload instruction to run the code.

Additional comments

The topic of the paper is interesting and relevant but not well organised in terms of presentation. Several typos and grammar mistakes are available and require a proper proofread. Also, please work more on the formatting of the paper.

Also, I don't know why the authors removed all figures and table at the end of the article. It's better to post at the same place where you gave a reference. The current view of the paper seems like essay and therefore not interesting for technical and quick readers.

---

## Round 0.2 · accepted · Accept

The authors have made all the required changes.

Reviewer 1 ·

Basic reporting

Authors added several figures and make the whole paper readable by general audience.

Experimental design

As I mentioned in previous major review, experimental design is ok.

Validity of the findings

Validity of finding is ok.

Additional comments

Authors improved the paper substantially and added several improvements and therefore no more further comments. Paper can be accepted in its present form.

Reviewer 2 ·

Basic reporting

Authors proposed all changes as mentioned in previous review report.

Experimental design

The experimental design of the paper is good, and it presents important results. No further comments.

Validity of the findings

Instructions are added for verifying code. No further comments.

Additional comments

Authors made all changes and presentation of the paper seems good and therefore paper can be accepted in present form.